# Fertilization and seasonality influence on the photochemical performance of tree legumes in forest plantation for area recovery in the Amazon

Roberto Kirmayr Jaquetti[1]*, José Francisco de Carvalho Gonçalves[1], Henrique Eduardo Mendonça Nascimento[1], Karen Cristina Pires da Costa[2], Jair Max Furtunato Maia[3], Flávia Camila Schimpl[4]

1 Laboratory of Plant Physiology and Biochemistry, National Institute for Amazonian Research, Manaus, Amazonas, Brazil, 2 Faculty of Agronomy, Institute of Studies in Agrarian and Regional Development, Federal University of South and Southeast of Pará, Marabá, Para, Brazil, 3 Biological Science Program, University of Amazonas State, Manaus, Amazonas, Brazil, 4 Federal Amazonian Institute, Presidente Figueiredo, Amazonas, Brazil

* jaquettiroberto@gmail.com

**Data Availability Statement:** The data underlying this study are available on Figshare (https://doi.org/

## Abstract

N-fixing leguminous species can reach atmospheric dinitrogen gas ($N_2$), having an advantage under N-limited degraded environments. These N-fixers are constantly used as facilitative species. Chlorophyll *a* fluorescence (ChF) acknowledges how different species take up and use light energy during photosynthesis. These techniques assess stress and performance responses to photosynthesis and are used for the selection of species with potential for reforestation. Six Fabaceae species were selected for this study: three nonfixing species (*Cenostigma tocantinum*, *Senna reticulata* and *Dipteryx odorata*) and three N-fixing species (*Clitoria fairchildiana*, *Inga edulis* and *Acacia* spp.). Variations in chlorophyll fluorescence under high vs. low water and nutrient conditions were studied. Multivariate analysis was performed to detect the effects of seasonality and fertilization on dark-adapted ChF two years after the experiment was established. The correlation among ChF variables and growth, photosynthesis and foliar nutrient concentrations was evaluated. Under high water- and nutrient-availability conditions, plants exhibited an enhanced performance index on absorption basis values correlated with electron transport fluxes. Under drought and nutrient-poor conditions, most species exhibit increased energy dissipation as photoprotection. High interspecific variation was found; therefore, species-specific responses should be considered in future ChF studies. Corroborating the ability to colonize high-light environments, N-fixers showed an increased performance index correlated with electron transport and Zn and N foliar concentrations. Negative correlations were found between photosynthesis and trapped fluxes. Diameter growth was positively correlated with electron transport fluxes. Given the different responses among species, ChF is an effective technique to screen for seasonality, fertilization and species effects and should be considered for use during forest restoration. Finally, the addition of fertilization treatments may facilitate tropical forest restoration due to the importance of nutrients in physiological processes. N-fixers showed high

10.6084/m9.figshare.13130279.v3 and https://doi.org/10.6084/m9.figshare.12869960.v6).

**Funding:** This work was financially supported by the: JFCG - Project Pro-Amazonia N°52 AUXPE 3390/2013 - Coordination for the Improvement of Higher-Level Personnel (CAPES, Brazil). https://www.gov.br/capes/pt-br JFCG - Universal 480233/2011-0 - National Council for Scientific and Technological Development (CNPq, Brazil). http://www.cnpq.br The funders had no role in study design, data collection and analysis, decision to publish, or preparation of the manuscript.

**Competing interests:** The authors have declared that no competing interests exist.

photochemical performance and tolerance to abiotic stress in degraded areas and therefore should be included to support ecosystem biomass restoration.

## Introduction

Plant species use light from the sun as their primary source of energy for photosynthesis, and physiologists currently want to understand how light interacts with the complex canopy of plants [1]. However, the excessive light energy found in degraded tropical areas (above 2000 µmol m$^{-2}$ s$^{-1}$ under natural conditions) may cause photoinhibition of late-successional species, affecting growth [2–4]. Under high-light environments, species capable of enhancing energy uptake and use will provide a desirable characteristic of individuals during tropical forest restoration [5, 6].

Fabaceae (legume) tree species have great ecological relevance during tropical forest regeneration [7]. From the six selected species, the nonfixing *Cenostigma tocantinum* Ducke and *Dipteryx odorata* (Aubl.) Willd. are associated with latter stages of succession. The nonfixing *Senna reticulata* (Willd.) H.S. Irwin & Barneby is a pioneer species common in floodplains in Amazonia [8]. N-fixing root nodule symbioses evolved in legumes, granting the ability to maximize nitrogen (N) uptake [9]. The inclusion of N-fixers may lower the external inputs and the time to rebuild C stocks [10]. The two N-fixers *Inga edulis* Mart. and *Acacia* spp. are well-known species used for facilitative purposes [11, 12]. *Clitoria fairchildiana* R.A. Howard is a less studied Amazonian N-fixer with great potential for use in tropical forest restoration.

Dark-adapted chlorophyll *a* fluorescence (ChF) measurements based on dissipative (*DI*), absorbed (*ABS*), trapped (*TR*) and transport (*ET*) energy fluxes are considered effective indicators of the effects of stress on photosynthetic performance [13, 14]. The OJIP curve (shape) has also been applied in the early detection of different abiotic stresses [15]. However, contrasting drought effects on the quantum yield of PSII photochemistry ($F_V/F_M$) and the performance index on an absorption basis ($PI_{ABS}$) have been reported on crop and tree species [16]. These results depend on the water deficit and the adjustments of electron transport and dissipation fluxes [17, 18].

Degraded areas in the Amazon Basin with decreased soil organic matter impose multiple nutrient limitations restricting photochemical activities and plant growth [19, 20]. Studies have demonstrated the effects of fertilization on biomass growth during forest restoration with leguminous trees [21, 22]. Fertilization effects may enhance the drought tolerance of species due to increased quantum yield [23, 24]. A few studies have evaluated the combined effects of seasonality and fertilization on chlorophyll fluorescence. A recent study with Brazil nut trees (*Bertholletia*), an adapted species to nutrient-poor soils, showed ChF adjustments to light environments when thinned and fertilized with phosphate [25, 26].

There has been increased interest in ChF responses to environmental cues and their correlation to photosynthesis and growth [27–29]. ChF has been demonstrated to be an effective technique to detect different environmental cues, although the large number of variables based on *ABS*, reaction centers (*RCs*) and cross-sections (*CSs*) may confuse researchers [30]. Multivariate analyses such as principal component analyses (PCAs) may reduce the noise associated with large ChF datasets, allowing researchers to understand how different species and ecological groups adjust energy fluxes [31]. Likewise, in addition to being less expensive than photosynthesis measurements, ChF can also detect effects such as seasonality, which are harder to notice during growth analysis under field conditions.

Understanding ChF responses in ecological restoration species may increase forest restoration success. While a substantial number of ChF studies exist, some knowledge gaps remain. For instance, no physiological studies on *C. fairchildiana* or *C. tocantinum* species have been published. Studies of the combined effects of fertilization and seasonality on ChF traits in

tropical tree legumes are also lacking. Studies have demonstrated the correlations between ChF variables and growth, photosynthesis and nutrient traits in *Bertholletia* and crop species [26, 32–34]. Therefore, three main hypotheses were tested: 1. Late-successional species will be phothoinhibited when planted in open areas, while N-fixing species perform well when fertilized. 2. How leguminous trees adjust their photochemical energy fluxes to different water and nutrient availability; and 3. ChF variables can be used as proxies to screen photosynthesis and growth during forest restoration.

## Materials and methods

### Experimental trial, species and treatments

Substantial variability in precipitation, temperature and soil properties can be found across Amazonian ecosystems. Central Amazonia is characterized by low natural soil fertility and high irradiance, temperatures, and monthly precipitation [35, 36]. Large- and small-scale degradation is widespread throughout the Amazon basin [37–39]. Detailed information about monthly precipitation and soil chemical characteristics in the experimental area can be found in [40]. The experimental trial was established in a typical homogeneous area of the Forest Restoration Program of the Balbina Hydropower Dam (1˚55'36"S, 59˚27'10"W) distant 145 km straight line from Manaus in Amazonas state, Brazil. The Balbina Hydropower Dam administration concede permission to establish, collect and use data from the experimental trial. Specific environmental or scientific permissions was not required for the experimental activities as no endangered or protected species were considered. The 3 ha area was degraded by the removal of the natural nonflooded and dense *terra firme* forest without burning. Despite maintaining its soil structure quality, nutrient deficiencies for N (0.16 g $kg^{-1}$), phosphorus (P) (0.14 mg $dm^{-3}$), calcium (Ca) and magnesium (Mg) (0.3 $cmol_c$ $dm^{-3}$) were found [35, 40]. With the bare soil, no natural regeneration of gramineas or pioneer species was observed in the area 30 years after abandonment, supporting the planting seedling choice. Nine blocks (n = 9) measuring 6 m x 72 m (432 $m^2$ area) each were placed across the area containing all 12 treatments in a combination of six tree legume species under low- (unfertilized) and high-nutrient (fertilized) treatments (Fig 1). The 108 studied plants were selected from a total of 432 individuals planted. The high-nutrient treatment received four applications of macro- and micronutrients at the beginning of the dry and wet seasons, while the low-nutrient treatment received no fertilization throughout the 24-month experiment.

Under normal conditions precipitation around the area is well distributed with less rainy months (approximately 150 mm $month^{-1}$) from June to November [40]. During the experimental course, the strong 2015/16 El Niño-Southern Oscillation (ENSO) event caused a 60-day period with no precipitation when the first measurements were performed (dry season). Second measurements were accomplished 6 months later (rainy season) after recovery.

The relative growth rate of the diameter ($RGR_D$) of each tree, hereafter the diameter growth, was calculated 24 months after planting [21]. More information on the species used, fertilization methods, photosynthetic pigment analysis and monthly precipitation during the experimental period can be found in [41].

### Photosynthesis measurements

Photosynthesis was measured between 8:00 and 11:30 h in nine selected plants per treatment in the dry and wet seasons. Healthy, sun-exposed and completely expanded leaves on the east side of the plants were selected from the middle third of each plant. The net photosynthetic rate ($P_n$), hereafter referred to as photosynthesis, was measured using a portable photosynthesis system (Li-6400, Li-Cor Inc., Lincoln, NE, USA) as described by [42]. Each measurement was performed at photosynthetic photon flux densities of 2000 μmol $m^{-2}$ $s^{-1}$ with the foliar

chamber adjusted to a $CO_2$ concentration, temperature and water vapor concentration of $400 \pm 4$ µmol mol$^{-1}$, $31°C \pm 1°C$ and $21 \pm 1$ mmol mol$^{-1}$, respectively.

## Chlorophyll *a* fluorescence

The dark-adapted ChF measurements were performed with a portable fluorometer (Handy PEA, MK2-9600-Hansatech, Norfolk, UK) on the same individuals and leaves used for the photosynthesis measurements between 8:30 and 11:00 h. The selected leaves were dark-adapted for a period of 30 min and then exposed to a 1 s excitation pulse of intensely saturating light (3000 µmol m$^{-2}$ s$^{-1}$) at a wavelength of 650 nm. Fast fluorescence transients were calculated based on the so-called "JIP-test" [43]. The $PI_{ABS}$, hereafter the performance index, which combines *DI*, *TR* and *ET* fluxes, was calculated according to Eq 1 described by [44]. The units and formulae used are provided in Table 1.

$$\frac{RC}{ABS} \times \left[\frac{\varphi P_0}{1 - \varphi P_0}\right] \times \left[\frac{\psi E_0}{1 - \psi E_0}\right] \qquad \text{Eq 1}$$

where *RC* = active PSII reaction centers, *ABS* = photon flux absorbed by the PSII antenna, $\varphi P_0$ = $F_V/F_M$ = maximum quantum yield of PSII photochemistry, $\psi E_0 = ET_0/TR_0$ = efficiency at which trapped electrons are transferred from $Q_A^-$.

## Foliar nutrient concentration

The macro- and micronutrient concentrations of the leaves were determined in each treatment after the leaf samples were oven-dried at 65°C and ground. The N concentration (mass basis)

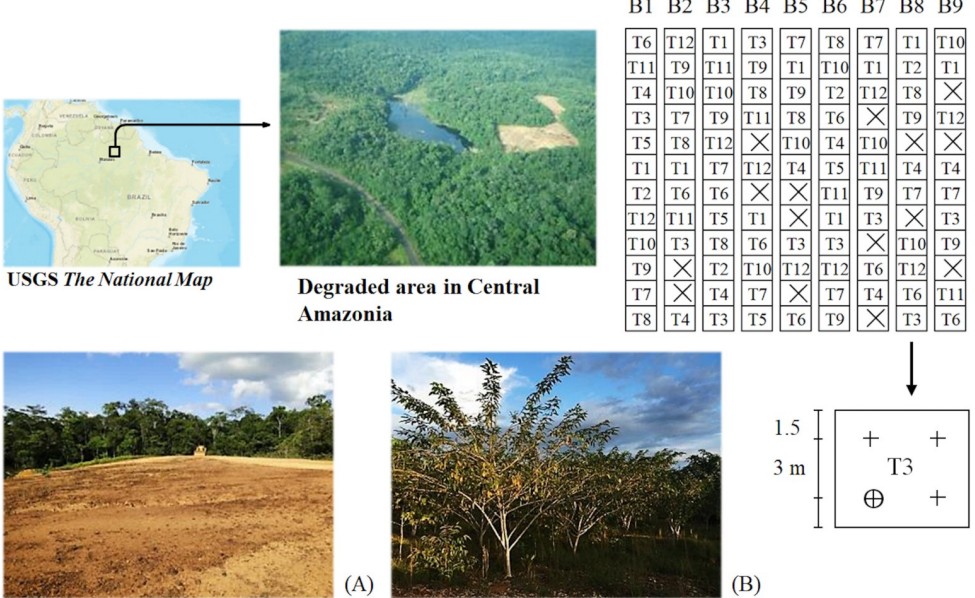

**Fig 1.** Graphical depiction with the location of the *"Alalau"* degraded area, the experimental design, and the apparent conditions before intervention (A) and 2 years after tropical forest restoration (B). Experimental design with 12 treatments (fertilization x species) randomly allocated in nine blocks (n = 9). Lost parcels (n) due to the mortality of four replicates are represented by the crossed squares (☒). T1 and T7, *Cenostigma tocantinum*; T2 and T8, *Senna reticulata*; T3 and T9, *Dipteryx odorata*, T4 and T10, *Clitoria fairchildiana*; T5 and T11, *Inga edulis*; T6 and T12, *Acacia* spp. T1 to T6, low-nutrient treatments; and T7 to T12, high-nutrient treatments. For example, the four replicates of the T3 treatment (low-nutrient x *D. odorata*) in block eight are defined with the cross symbol (✚) in the enlarged parcel, and the circulate cross (⊕) illustrates the selected plant for data sampling. The map figure was sourced from United States Geological Survey—USGS National Map Viewer (http://viewer.nationalmap.gov/viewer/).

**Table 1. ChF variable meanings, units and equations used.**

| Variables | Meaning | Units and equations |
|---|---|---|
| $F_0$ | Initial fluorescence | Fluorescence intensity at 0.05 ms |
| $F_M$ | Maximum fluorescence | Fluorescence intensity at 30 ms |
| $F_V$ | Maximum variable fluorescence | $(F_M - F_0)$ |
| $F_V/F_M$ | Maximum quantum yield of PSII photochemistry | $\varphi P_0 = \frac{TR_0}{ABS} = \frac{F_V}{F_M}$ |
| $PI_{ABS}$ | Performance index on an absorption basis | $\frac{RC}{ABS} \times \left[\frac{\varphi P_0}{1 - \varphi P_0}\right] \times \left[\frac{\psi E_0}{1 - \psi E_0}\right]$ |
| $ET_0/TR_0$ | Efficiency of electron transport after $Q_A^-$ | $\psi E_0 = 1 - V_j$ |
| $TR_0/RC$ | Maximum trapped exciton flux of an active PSII | $\left(\frac{M_0}{V_j}\right)$ |
| $ABS/RC$ | Apparent antenna size of an active PSII | $\left(\frac{M_0}{V_j}\right)/\varphi P_0$ |
| $ET_0/RC$ | Electron transport flux per active PSII | $\left(\frac{TR_0}{RC}\right) \times \psi E_0$ |
| $DI_0/RC$ | Energy dissipation flux per active PSII | $\left(\frac{ABS}{RC}\right) - \left(\frac{TR_0}{RC}\right)$ |
| $RC/CS$ | Density of reaction centers per cross-section | $\left(\frac{ABS}{CS}\right) \times \left(\frac{RC}{ABS}\right)$ |

was determined with a 2400 Series II CHNS/O Organic Elemental Analyzer (PerkinElmer Inc., Waltham, MA, USA). The P concentration was determined by spectrophotometry at 725 nm. The potassium (K), Ca and Mg, iron (Fe) and zinc (Zn) concentrations were determined using atomic absorption spectrophotometry (PerkinElmer 1100B, Inc., Waltham, MA, USA) according to dos Santos Junior [42].

## Data analysis

A complete randomized block experimental design was used. The interrelationships among ChF variables were assessed using the PCA ordination method, which reduces the dimensionality of the original data [45]. PCA was performed to evaluate the effects of the seasonality and fertilization treatments. All variables were standardized by the maximum relatedness [46] prior to analysis. Product-moment correlations were used to assess the influences of seasonality (dry and wet seasons) and fertilization (fertilized and unfertilized treatments) on the ordination axes and each original variable. At a probability level of $P < 0.05$, pairwise t-tests were performed to evaluate the significance of the seasonality and fertilization effects. The analyses were run initially with 21 ChF variables and were run again with the 11 most responsive variables after removing similar and nonresponsive variables (Table 2).

The effects of seasonality and fertilization on the performance index and the energy dissipation flux per active PSII ($DI_0/RC$), hereafter the energy dissipation, were compared using repeated measure two-way ANOVA with seasonality (dry and wet) as the repeated measure and species and fertilization treatments as factors. The relationships among ChF variables and functional traits were tested using nonparametric Spearman pairwise correlation analysis in the fertilized plants and during the wet season. PCA was performed with PAST-UiO 3.0 (Hammer and Harper, Oslo, NO), and inferential tests were performed with STATISTICA 12.0 (TIBCO Software Inc., CA, USA).

## Results

### Seasonal effects in the high- and low-nutrient treatments

Corroborating the first hypothesis, plants in the high-nutrient treatment adjust their energy fluxes to seasonality. Significant differences separating dry and wet seasons were found in PCA

**Table 2. Effects of seasonality and fertilization with 21 and 11 ChF and 5 photosynthetic pigment variables.**

| Variables arrangement | Seasonality effects | | | | | | | |
|---|---|---|---|---|---|---|---|---|
| | High-nutrient | | | | Low-nutrient | | | |
| | PCA 1 | | PCA 2 | | PCA 1 | | PCA 2 | |
| | $P$ | t-test | $P$ | t-test | $P$ | t-test | $P$ | t-test |
| 21 ChF + 5 Chl | <0.01 | -3.20 | <0.08 | -1.75 | <0.05 | -2.18 | 0.44 | -0.77 |
| 11 ChF + 5 Chl | <0.001 | -3.62 | 0.11 | -1.61 | <0.05 | -2.29 | 0.42 | -0.82 |
| 11 ChF | <0.01 | -3.12 | 0.10 | -1.68 | <0.05 | -2.05 | 0.41 | -0.83 |
| | Fertilization effects | | | | | | | |
| | Wet season | | | | Dry season | | | |
| | PCA 1 | | PCA 2 | | PCA 1 | | PCA 2 | |
| | $P$ | t-test | $P$ | t-test | $P$ | t-test | $P$ | t-test |
| 21 ChF + 5 Chl | <0.01 | -3.12 | <0.01 | -2.74 | <0.01 | -3.72 | 0.07 | -1.85 |
| 11 ChF + 5 Chl | <0.001 | -3.59 | <0.05 | -2.04 | <0.001 | -4.44 | 0.35 | -0.94 |
| 11 ChF | <0.01 | -3.14 | <0.05 | -2.59 | <0.001 | -3.61 | 0.21 | -1.26 |

axis 1 (t = -3.12, $P$ < 0.01). Reducing from 21 to 11 ChF variables enhanced the seasonality differences according to $P$ and t values (Table 2). Most individuals enhanced the energy dissipation fluxes and apparent antenna size of an active PSII ($ABS/RC$), hereafter antenna size, in the dry season. Enhanced efficiency of electron transport after $Q_A^-$ ($ET_0/TR_0$), hereafter electron transport, and performance index were found in the wet season (Fig 2A). Low-nutrient plants adjust poorly to seasonal effects with no clear separation between dry and wet seasons (Fig 2B). Slight differences were found in PCA axis 1 (t = -2.05, $P$ < 0.05) (Table 2), suggesting that fertilization treatment enhanced leguminous tree tolerance to dry periods through ChF adjustments.

## Fertilization effects during dry and wet seasons

Most plants adjust their energy fluxes to different nutrient regimes during the wet season. An evident separation between high- and low-nutrient treatments was found in PCA axis 1 (t = -3.14, $p$ < 0.01) (Table 2). Plants in the high-nutrient treatment exhibited higher performance index values and greater electron transport, while low-nutrient plants exhibited increased energy dissipation and initial Chl $a$ fluorescence ($F_0$), hereafter the initial fluorescence (Fig 3A). In the dry season, a noticeable effect for the fertilization treatment was found in PCA axis 1 (t = -3.61, $p$ < 0.001). Low-nutrient plants contributed to increased energy dissipation fluxes, while high-nutrient plants improved electron transport and the electron transport flux per active PSII ($ET_0/RC$) (Fig 3B). These findings suggest that the drought tolerance of leguminous trees increased under the fertilization treatments. Low-nutrient *Acacia* spp. adjusted ChF variables differently with high electron transport and performance index values independent of seasonal effects (S1 Table).

## Seasonal, fertilization and specific effects on $PI_{ABS}$ and $DI_0/RC$

Considering the performance index, significant effects were found for the fertilization and species treatments with a significant interaction between factors. For the repeated measures, differences were found between dry and wet seasons with the interaction between species and seasonal effects. No interaction was found between seasonal and fertilization effects (S2 Table). When species were compared, significant differences were found between N-fixing and non-fixing species, with the highest performance values for *Acacia* spp. and *I. edulis* (Fig 4A).

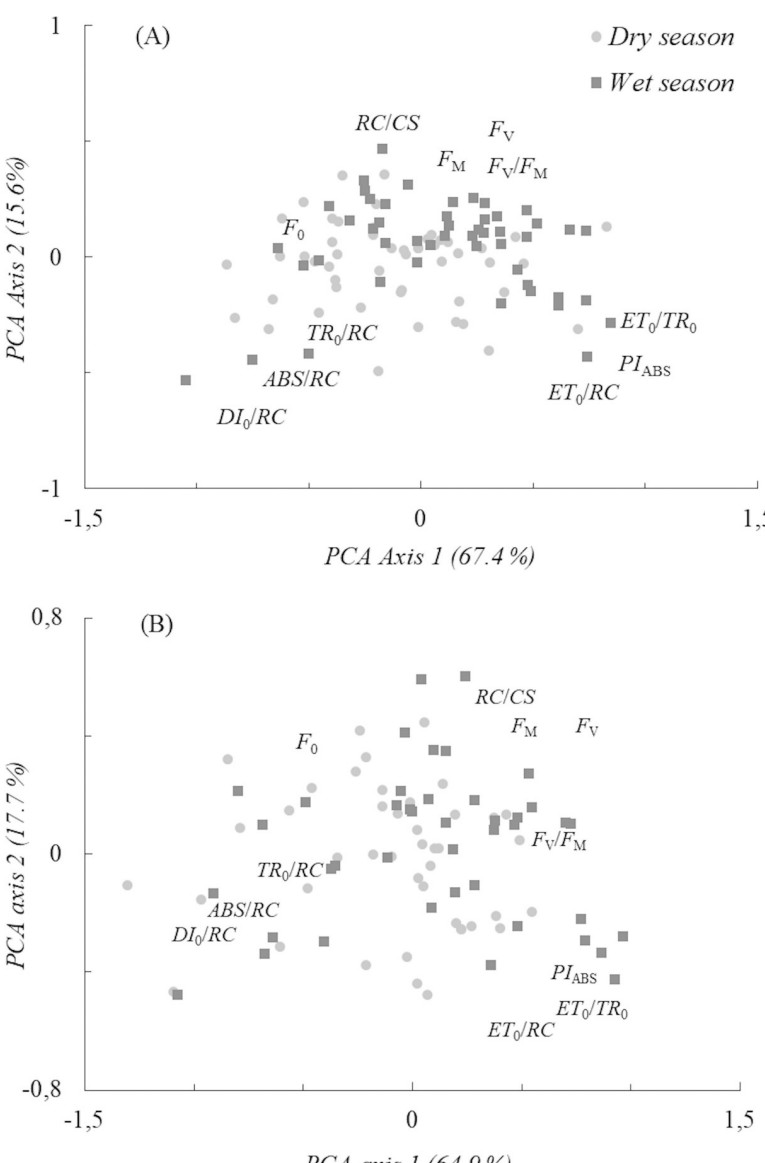

**Fig 2.** PCA ordination of seasonal effects in the high- (A) and low-nutrient (B) treatments. $F_0$, initial, $F_M$, maximum and $F_V$, variable fluorescences; $F_V/F_M$, maximum quantum yield; $PI_{ABS}$, performance index; $ET_0/TR_0$, efficiency of electron transport; $DI_0/RC$, energy dissipation; $ABS/RC$, antenna size; $ET_0/RC$, electron transport flux; $TR_0/RC$, maximum trapped exciton flux; $RC/CS$, density of reaction centers per cross-section. ChF variables are approximately placed according to the product-moment (correlation loading) values along both PCA axes.

Considering the energy dissipation, significant differences were found for both fertilization and species treatments with interactions between factors. For the seasonal effects, significant differences were found among dry and wet seasons with no interactions among factors. Comparing the species, the late-successional *D. odorata* increased energy dissipation, while no differences were found among other species ([Fig 4B]).

## Correlation of ChF with photosynthesis, foliar nutrients and plant growth

Under high water and nutrient availability, the performance index was positively correlated with diameter growth ($r_s = 0.65$) and Zn ($r_s = 0.52$). The maximum trapped exciton flux of

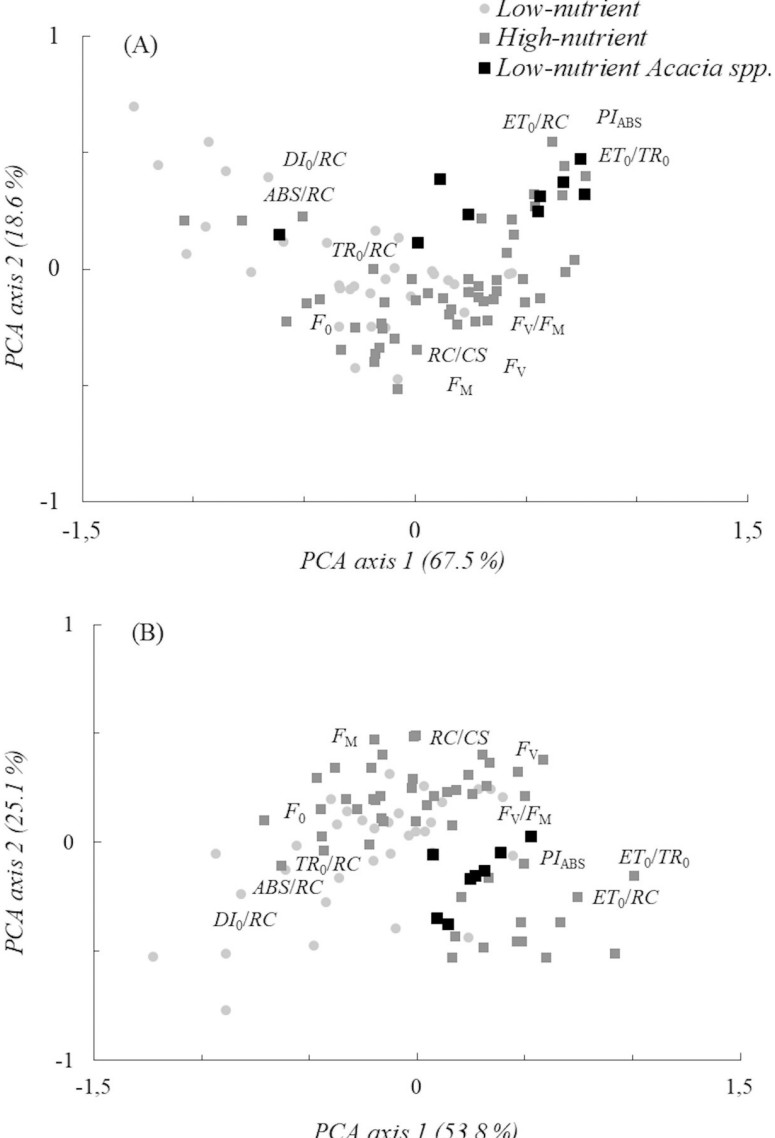

**Fig 3.** PCA ordination of fertilization effects during the wet (A) and dry (B) seasons. $F_0$, initial, $F_M$, maximum and $F_V$, variable fluorescences; $F_V/F_M$, maximum quantum yield; $PI_{ABS}$, performance index; $ET_0/TR_0$, efficiency of electron transport; $DI_0/RC$, energy dissipation; $ABS/RC$, antenna size; $ET_0/RC$, electron transport flux; $TR_0/RC$, maximum trapped exciton flux; $RC/CS$, density of reaction centers per cross-section. ChF variables are approximately placed according to the product-moment (correlation loading) values along both PCA axes.

active PSII ($TR_0/RC$), hereafter the trapped flux, was negatively correlated with photosynthesis ($r_s$ = -0.66), N ($r_s$ = -0.58) and Zn ($r_s$ = -0.53). Negative correlations with moderate collinearity were found between diameter growth and $F_0$ ($r_s$ = -0.61) and energy dissipation ($r_s$ = -0.57) (Table 3). Electron transport was positively correlated with Zn ($r_s$ = 0.51) and growth ($r_s$ = 0.63) and negatively correlated with Ca ($r_s$ = -0.58).

## Fertilization effects on OJIP transient curves

Nutrient deficiencies affected the curvature of the OJIP transient curves, reducing the slope for most species. *C. tocantinum* was greatly affected during the J-I rise, while *S. reticulata* reduced

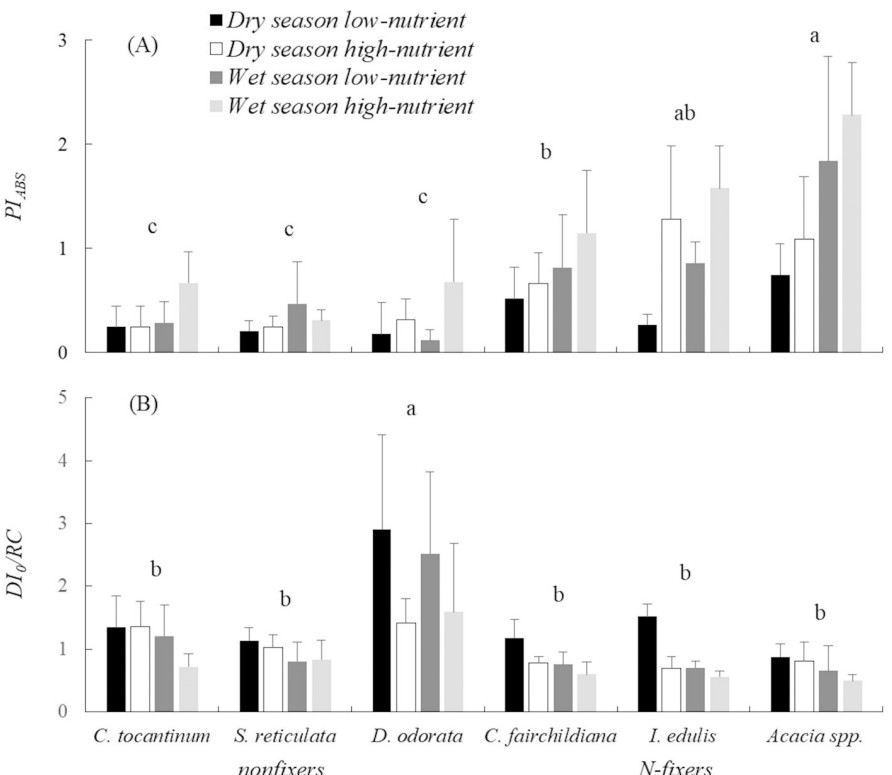

**Fig 4.** Performance index (A) and energy dissipation (B) of the six studied species under low vs. high water and nutrient availability (mean values ± SD).

the O-J and I-P rise. Unfertilized *C. fairchildiana* had a general reduction in the transient rise (Fig 5). *I. edulis* appears less sensitive to OJIP modifications. Fertilized *D. odorata* reduced the O-J-I rise with increased $F_0$ and reduced maximum Chl *a* fluorescence ($F_M$), hereafter the maximum fluorescence (Fig 5). The enhanced photochemical efficiency of *Acacia* spp. was due

**Table 3. Pairwise Spearman correlation coefficients ($r_S$) among ChF variables and growth, photosynthesis and foliar nutrient concentrations.**

|  | $RGR_D$ | $P_n$ | N | P | K | Ca | Mg | Fe | Zn |
|---|---|---|---|---|---|---|---|---|---|
| $F_0$ | **-0.61**\*\* | -0.35\*\* | -0.28\* | 0.00$^{ns}$ | 0.21$^{ns}$ | 0.11$^{ns}$ | -0.14$^{ns}$ | -0.28\* | -0.26$^{ns}$ |
| $F_M$ | 0.31$^{ns}$ | -0.05$^{ns}$ | 0.08$^{ns}$ | 0.34\* | -0.25$^{ns}$ | 0.41\*\* | 0.02$^{ns}$ | -0.19$^{ns}$ | -0.09$^{ns}$ |
| $F_V$ | -0.13$^{ns}$ | 0.05$^{ns}$ | 0.18$^{ns}$ | 0.33\* | -0.30\* | 0.37\*\* | 0.08$^{ns}$ | -0.10$^{ns}$ | 0.01$^{ns}$ |
| $F_V/F_M$ | **0.51**\*\* | 0.29$^{ns}$ | 0.28\* | 0.08$^{ns}$ | -0.27$^{ns}$ | 0.03$^{ns}$ | 0.11$^{ns}$ | 0.24$^{ns}$ | 0.22$^{ns}$ |
| $PI_{ABS}$ | **0.65**\*\* | 0.40\* | 0.49\*\* | -0.21$^{ns}$ | -0.17$^{ns}$ | -0.40\*\* | -0.02$^{ns}$ | 0.37\*\* | **0.52**\*\* |
| $ET_0/TR_0$ | **0.63**\*\* | 0.38\* | 0.46\*\* | -0.32\* | -0.17$^{ns}$ | **-0.50**\*\* | -0.08$^{ns}$ | 0.33\* | **0.51**\*\* |
| $TR_0/RC$ | -0.44\*\* | **-0.66**\*\* | **-0.58**\*\* | -0.03$^{ns}$ | 0.09$^{ns}$ | -0.41\*\* | 0.08$^{ns}$ | -0.26$^{ns}$ | **-0.53**\*\* |
| $DI_0/RC$ | **-0.57**\*\* | -0.44\*\* | -0.42\*\* | -0.04$^{ns}$ | 0.23$^{ns}$ | 0.11$^{ns}$ | -0.08$^{ns}$ | -0.26$^{ns}$ | -0.35\* |

$F_0$, initial fluorescence; $F_M$, maximum fluorescence; $F_V$, maximum variable fluorescence; $F_V/F_M$, maximum quantum yield of PSII photochemistry; $PI_{ABS}$, performance index on an absorption basis; $ET_0/TR_0$, efficiency of electron transport; $TR_0/RC$, maximum trapped exciton flux; $DI_0/RC$, energy dissipation flux; $RGR_D$, relative growth rate in diameter; $P_n$, net photosynthetic rate; N, nitrogen, P, phosphorous, K, potassium, Ca, calcium, Mg, magnesium, Fe, iron and Zn, zinc concentrations in leaves. Bold values are highly significant ($p < 0.001$) with high ($r_s > 0.70$) and moderate ($0.50 < r_s < 0.70$) collinearity between variables.

\* Significance at the 0.05 level

\*\* Significance at the 0.01 level.

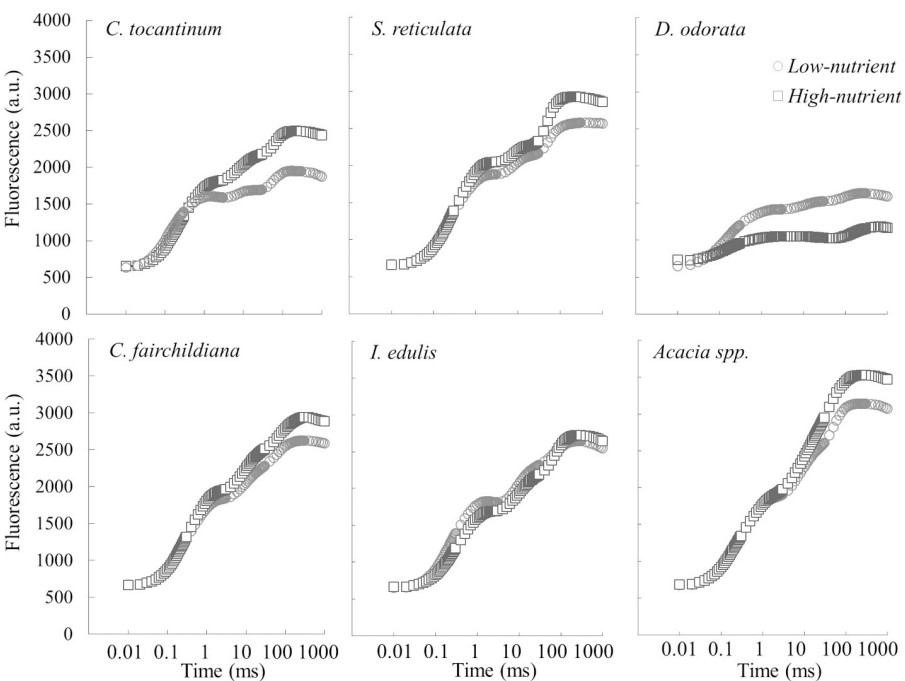

**Fig 5. The OJIP transient curve for the six studied species of high- and low-nutrient plants.**

to the increased $F_M$ values and the absence of the P step. The tolerance of *Acacia* spp. to nutrient deficiencies was supported by the enhanced transient rise and the presence of the I and P steps.

## Discussion

### Specific performance and stress responses

Photoinhibition responses were found in late-successional *D. odorata* with increased energy dissipation. On the other hand, the high performance of N-fixers, especially *Acacia* spp., was evidenced under different water and nutrient conditions.

As hypothesized, we expect that late-successional species may be photoinhibited when exposed to excessive light energy in degraded areas [47, 48]. Late-successional *D. odorata* increased energy dissipation, indicating photoinhibition and oxidative stress [13]. The increased initial fluorescence and decreased maximum fluorescence in the *D. odorata* curve suggest photoprotective responses through RC inactivation [49, 50]. The late-successional *C. tocantinum* appeared to be less photoinhibited under high-light conditions. Increased energy dissipation under high light has been reported for tropical tree species [4, 43, 51].

The reduced J-I-P rise found in nonfixers (Fig 3) reflected stress responses with decreased activity of PSII, accumulation of reduced $Q_A^-$ and impaired electron transport [52, 53]. Considering the performance index, nonfixers, including *S. reticulata* species, performed poorly compared to N-fixing species. The enhanced performance of N-fixers (*C. fairchildiana*, *I. edulis* and *Acacia* spp.) was partially explained by the increased electron transport and N concentration in leaves. The increased electron transport and I-P rise in *Acacia* spp. may be related to the reduction rate of PSI acceptor side transporters [52].

Collectively, our results reveal that ChF study can be used to screen the stress and performance responses of different species during forest restoration. ChF was less effective in

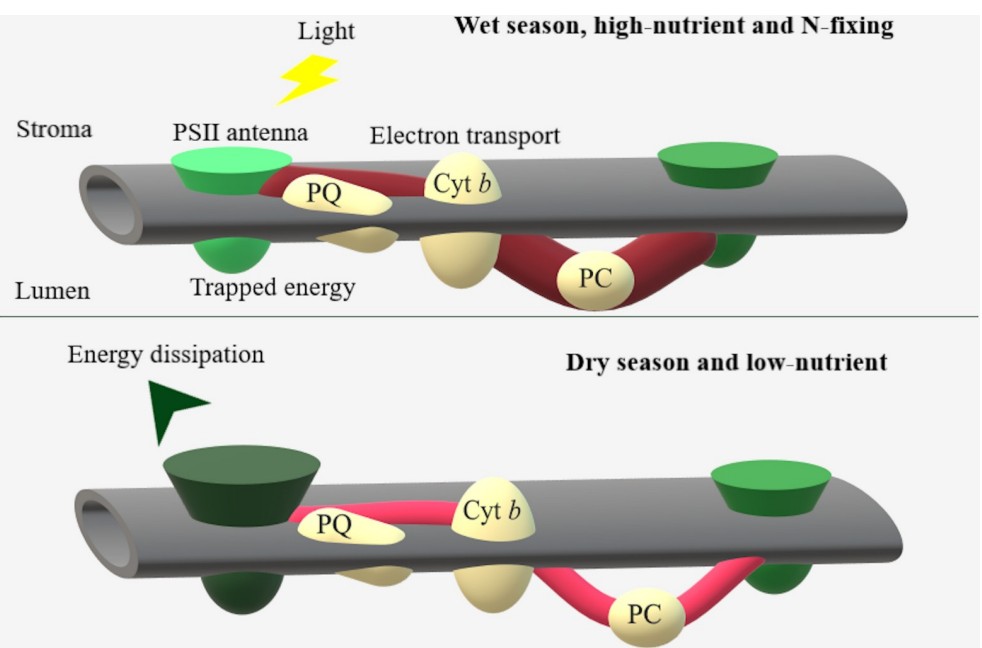

**Fig 6. ChF adjustments in antenna size, energy dissipation and electron transport.** Dark green and red indicate increased values of PSII antenna size and electron transport, and light green and red indicate decreased values.

distinguishing ecological groups where specific responses should be considered. N-fixers performed better than nonfixers, confirming the recommendation of use in degraded areas.

## Seasonal and nutrient availability effects on ChF

As general responses of the studied species, plants under low resources exhibited enhanced energy dissipation and antenna size. In contrast, under high water and nutrient availability, plants increased their electron transport and performance index (Fig 6).

With high water and nutrient availability, plant energy is mainly directed to photochemistry, thus improving photosynthetic performance [52, 54]. The reduced electron transport and performance under drought and nutrient deficiencies negatively affected photosynthesis and growth. Extreme climatic events such as ENSO may increase dry periods in certain regions of the Amazon Basin [55]. After downregulating photosynthesis, photochemistry and electron transport may be affected under moderate drought conditions [56, 57]. By maintaining electron transport longer *Acacia* spp. species can increase the tolerance to dry periods [58, 59]. Moreover, in species of the *Acacia* genus, negligible effects on quantum yield were found even under severe water stress [60].

The increased tolerance to dry periods found under fertilization in some species in the present study may be related to the increased electron transport and photochemical efficiency and photoinhibition relief [20, 61]. Plants may respond to fertilization treatments with increased plant performance and photosynthesis due to the importance of nutrients in physiological processes [54, 62]. Through adjusting the N concentration, gas exchange and growth, N-fixers may take up and use resources efficiently, particularly in nutrient-limited soils [54, 63]. The diminished electron transport reported under nutrient deficiencies may be partially due to the overreduction of the PSII acceptor side [61]. Multiple nutrient deficiencies reduce the slope of the OJIP curves in most species due to reduced ATP production under K and P deficiencies [15, 64].

Increased energy dissipation and antenna size have been previously reported as early indicators of drought and nutrient deficiency effects [15, 65]. Both positive and negligible effects of drought on PSII quantum yield have been previously reported; the effect depends on the nutrient status of individual plants [66, 67]. Corroborating previous findings, the performance index was a more sensitive variable for evaluating fertilization and drought effects than the PSII quantum yield [44, 68].

Here, multivariate analysis demonstrated a positive correlation between plant performance and electron transport adjustments. Our results indicate a positive effect of fertilization on enhancing the drought tolerance of species. Through electron transport adjustments, *Acacia* spp. can tolerate water and nutrient deficiencies longer and should be considered to restore highly degraded areas.

## Correlation among ChF variables, growth and photosynthesis

The present study demonstrates the negative correlations between trapped energy flux and photosynthesis and between initial fluorescence and growth. Photosynthesis was negatively correlated with trapped flux, while diameter growth was positively correlated with the performance index and electron transport. These findings have ecological and silvicultural importance for potential use to screen well-adapted species for degraded environments.

The $PI_{ABS,}$ which incorporates the density of PSII RCs, electron transport beyond $Q_A$, and trapped fluxes, is widely used to study photosynthesis and the functionality of PSII and PSI under stress [3, 16]. The increased performance index values found under high resource availability and in N-fixing species may result from changes in the PSII antenna size, trapping efficiency and electron transport [14, 43]. The electron transport flux variables reflect the maximum electron transport between PSII and PSI and indicate changes in photosynthetic apparatus activity [69].

Corroborating previous studies, the positive correlation between energy dissipation and the initial fluorescence of stressed individuals may be associated with increased antenna size and PSII RC inactivation [17]. The negative correlation between dissipative energy and $P_n$ suggests decreased electron transport and photochemical yield.

The quantum yield was positively correlated with growth, but no correlation was found with photosynthesis, contrasting with recent findings [70]. Supporting our hypothesis, a positive correlation was found among growth, electron transport and the performance index. These findings may have important applications to improve the growth and tolerance of individuals to stress. Moreover, the performance index appears to be a potential proxy for biomass growth and productivity during reforestation. The negative correlation between Ca and electron transport may be related to the secondary messenger functions of Ca [71]. A positive correlation was also found between leaf Zn concentrations and the performance index in *Bertholletia* species [26].

In addition to the present results, positive correlations were found among photosynthesis, quantum yield and maximum fluorescence in *Populus* and *Miscanthus* species [34, 72]. Positive correlations were previously found among Fe, the performance index and electron transport [26] and between electron transport and photosynthesis [32, 33]. In contrast to our results, no correlation was observed between photosynthesis and ChF variables in barley plants under drought and control treatments [18].

The negative correlation between the N concentration and trapping energy confirms the advantage of N-fixers in decreasing trapped flux and moving electrons beyond $Q_A$. Overall, the ChF technique was deemed effective in detecting seasonality and fertilization effects, although specific responses should be considered.

## Conclusions

The ChF technique is valuable for understanding the photochemical phase of photosynthesis and how it affects other functional traits. Nevertheless, ChF has rarely been used to assess the photosynthetic performance and adaptation ability of species used in forest restoration. As evidenced in the present study, adjustments in energy fluxes during light uptake are determinants of the growth and establishment of different species. Additionally, plants can increase energy dissipation when resources are scarce and enhance electron transport when resources are abundant. N-fixing species with enhanced performance appear to be highly adapted to degraded, high-light environments. In particular, the increased electron transport fluxes in *Acacia* spp. may explain the enhanced sink strength and growth of these species in locations with multiple resource limitations. Future studies on the physiological traits and quenching analysis of ChF of leguminous trees are recommended, especially on N-fixers that may facilitate the restoration of important biogeochemical cycles.

## Supporting information

**S1 Table. Mean ChF variable values for the six studied species under the different water and nutrient regimes.** $F_0$, initial fluorescence; $F_M$, maximum fluorescence; $F_V$, maximum variable fluorescence; $F_V/F_M$, maximum quantum yield of PSII photochemistry; $PI_{ABS}$, performance index on an absorption basis; $ET_0/TR_0$, efficiency of electron transport; $DI_0/RC$, energy dissipation flux; $ABS/RC$, antenna size of an active PSII $RC$; $ET_0/RC$, electron transport flux; $TR_0/RC$, maximum trapped exciton flux; $RC/CS$, density of reaction centers per cross-section. (DOCX)

**S2 Table. ANOVA results for the effects of seasonality, fertilization and species on $PI_{ABS}$ and $DI_0/RC$.** $PI_{ABS}$, performance index on an absorption basis; $DI_0/RC$, energy dissipation flux. Degrees of freedom for species = 5, and fertilization = 1. * Significance at the 0.05 level; ** Significance at the 0.01 level. (DOCX)

## Acknowledgments

The authors are grateful to the Balbina Hydroelectric Dam for enabling the collection of experimental data and to the editorial board for their valuable revisions.

## Author Contributions

**Conceptualization:** Roberto Kirmayr Jaquetti, José Francisco de Carvalho Gonçalves.

**Data curation:** Roberto Kirmayr Jaquetti, Karen Cristina Pires da Costa.

**Formal analysis:** Roberto Kirmayr Jaquetti, Henrique Eduardo Mendonça Nascimento.

**Funding acquisition:** José Francisco de Carvalho Gonçalves.

**Investigation:** Roberto Kirmayr Jaquetti, Karen Cristina Pires da Costa.

**Methodology:** Roberto Kirmayr Jaquetti, José Francisco de Carvalho Gonçalves, Karen Cristina Pires da Costa.

**Project administration:** Roberto Kirmayr Jaquetti.

**Resources:** José Francisco de Carvalho Gonçalves, Jair Max Furtunato Maia, Flávia Camila Schimpl.

**Software:** Roberto Kirmayr Jaquetti, Karen Cristina Pires da Costa.

**Supervision:** José Francisco de Carvalho Gonçalves.

**Validation:** José Francisco de Carvalho Gonçalves, Henrique Eduardo Mendonça Nascimento, Jair Max Furtunato Maia.

**Visualization:** Roberto Kirmayr Jaquetti.

**Writing – original draft:** Roberto Kirmayr Jaquetti, José Francisco de Carvalho Gonçalves.

**Writing – review & editing:** Roberto Kirmayr Jaquetti, Henrique Eduardo Mendonça Nascimento, Karen Cristina Pires da Costa, Jair Max Furtunato Maia, Flávia Camila Schimpl.

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
