## [Decision Letter · Decision Letter 0]

4 Jan 2021

PONE-D-20-35817

Effects of seasonality, fertilization and species on the chlorophyll a fluorescence as related with photosynthesis and leguminous tree growth during Amazonian forest restoration

PLOS ONE

Dear Dr. Jaquetti,

Thank you for submitting your manuscript to PLOS ONE. After careful consideration, we feel that it has merit but does not fully meet PLOS ONE’s publication criteria as it currently stands. Therefore, we invite you to submit a revised version of the manuscript that addresses the points raised during the review process.

As you will see, your manuscript has been accessed by two experts and although one has misunderstood the manuscript itself, I think that the comments raised by the second reviewer were overall highly positive and constructive. 

We look forward to receiving your revised manuscript.

Kind regards,

Wagner L. Araujo, Ph.D

Academic Editor

PLOS ONE

Journal Requirements:

2. We note that Figure 1 in your submission contain map images which may be copyrighted. All PLOS content is published under the Creative Commons Attribution License (CC BY 4.0), which means that the manuscript, images, and Supporting Information files will be freely available online, and any third party is permitted to access, download, copy, distribute, and use these materials in any way, even commercially, with proper attribution. For these reasons, we cannot publish previously copyrighted maps or satellite images created using proprietary data, such as Google software (Google Maps, Street View, and Earth). For more information, see our copyright guidelines: http://journals.plos.org/plosone/s/licenses-and-copyright.

2.1.    You may seek permission from the original copyright holder of Figure 1 to publish the content specifically under the CC BY 4.0 license. 

2.2.    If you are unable to obtain permission from the original copyright holder to publish these figures under the CC BY 4.0 license or if the copyright holder’s requirements are incompatible with the CC BY 4.0 license, please either i) remove the figure or ii) supply a replacement figure that complies with the CC BY 4.0 license. Please check copyright information on all replacement figures and update the figure caption with source information. If applicable, please specify in the figure caption text when a figure is similar but not identical to the original image and is therefore for illustrative purposes only.

Reviewers' comments:

Reviewer's Responses to Questions

**Comments to the Author**

1. Is the manuscript technically sound, and do the data support the conclusions?

Reviewer #1: Yes

Reviewer #2: Partly

2. Has the statistical analysis been performed appropriately and rigorously? 

Reviewer #1: Yes

Reviewer #2: N/A

3. Have the authors made all data underlying the findings in their manuscript fully available?

Reviewer #1: Yes

Reviewer #2: Yes

4. Is the manuscript presented in an intelligible fashion and written in standard English?

Reviewer #1: Yes

Reviewer #2: Yes

5. Review Comments to the Author

Reviewer #1: The authors present a good work characterizing the use of chlorophyll a fluorescence as a fundamental parameter for identification of trees with potential for reforestation in the Amazon forest. The authors used 6 leguminous tree species and performed a 24-month experiment, analyzing morphophysiological and nutritional characteristics in the dry and rainy seasons, in addition to treatments with high and low nutritional availability. The Abstract section needs to be rewritten to make the idea of the manuscript more evident, for instance, talking about which species were used (and why?) and about the aim of testing chlorophyll a fluorescence as a technique for selection and/or classification of species with potential for reforestation. The authors also need to reorganize the Introduction and write about which species were used and why. The Methodology is well presented, but crucial information about the location of the experiment and climatology is lacking. IMHO, most of the results at supplementary data should be presented in the body of the manuscript, in Results section but not only, because as currently form of the manuscript it is hard to interpret and understand the data. Furthermore, it is not clear which parameters were used to define whether data is supplementary or necessary for the manuscript. The Discussion is exceedingly difficult to follow regarding the species used in the experiment because they were not presented properly at the beginning of the Manuscript. Besides that, the results (figures and tables) need to be cited and highlighted in the Discussion section. The layout and presentation of Figures, Tables and Supplementary data need to be changed to improve the understanding of the manuscript.

Point-by-point revision:

Effects of seasonality, fertilization and species on the chlorophyll a fluorescence as related with photosynthesis and leguminous tree growth during Amazonian forest restoration

The title needs to be improved, for now it is too long and messy.

Abstract

L3-5: Remove “General and specific” and Rewrite. Merge these sentences. If these “conditions have been studied” what we know until now?

L8: Please rewrite this sentence.

L9: Remove “Additionally,”.

L14: What type of “performance index”?

L16: Change to “High”

L17-19: Why?

L21-23: If the work was (at least partially) a proof of concept of these "cost-effective technique" this information should be described in the beginning of Abstract.

L24: Remove “greatly”.

Introduction

L33-34: How these N2-fixing tree species enhance light uptake? What the links between soil nutrition and light assimilation? Pigment production? Explore this topic.

L38: “Various”? What effects? Describe these effects on photochemical components.

L39: What type of “performance index”?

L40: Which “tested species”? Trees? Crops? Native species? Model species?

L44-45: Reported in which species?

L61-62: Why these species are important?

L63: Describe and present these “few studies”.

L65: The authors need to talk more about "Leguminous trees" in the Introduction, which species were used in this work?

L66: Explore more the “Evolutionary N2-fixing species” and remove “will”.

Materials and methods

L71-73: Add some reference about climate characteristics, for instance: Alvares et al. 2013 (https://doi.org/10.1127/0941-2948/2013/0507).

L73-74: Remove “Without delving into geopolitical and scientific discussion”.

L74-75: Add references about “Large- and small-scale degradation” in Amazon basin, for instance: https://doi.org/10.1038/s41893-020-0492-y;
https://doi.org/10.1073/pnas.1905315116;
https://doi.org/10.1038/srep41489;
https://doi.org/10.1111/nyas.14364;
https://doi.org/10.1126/sciadv.aba2949;
https://doi.org/10.1038/nclimate3238;
https://doi.org/10.1038/s41467-017-02771-y

L77: “Located two hours by car from Manaus”? Please use distance in Kilometers and add the geographic coordinates.

L80: What means “good physical characteristics of soil”? Which one traits and what values?

L82: Where “Natural regeneration was not observed”? Describe better.

L87: Change “season” to “seasons”.

L105-107: This is Results and/or Discussion.

L113-115: This information needs to be easily accessed. Climatological data should be presented in the manuscript. Add this information at least as Supplemental Information.

L119: Why “east side”?

L122: Light curve?

L132: Demonstrate in the Methods how this “Performance Index” was calculated.

L133: “Table S2” should be present in the manuscript body, not as supplementary information.

L153: “S1 Table” is Results.

Results

L166: “S1 Table” should not be supplementary data and it must be presented in the body of the manuscript.

L168: S2 Table should not be supplementary data and it must be presented in the body of the manuscript.

L172: “S1 Table and S1 Fig” should not be supplementary data and it must be presented in the body of the manuscript.

L183: “S1 Table” should not be supplementary data and it must be presented in the body of the manuscript.

L192: “S3 Table”, it is OK this table be present in supplementary information.

L203: “S4 Table”, it is OK this table be present in supplementary information.

L225: “Table 1” should be “Table 3”.

Discussion

L241: It is important that the Discussion starts with a sentence from the authors, a summary of all results and initial conclusion.

L242: Add “availability” between “nutrient” and “effects”.

L261-264: How this novel finding is related with your main hypothesis?

L265-277: How these information of other authors is related with your findings in this work? Please, merge these paragraphs and add a partial conclusion.

L286: Please, add a conclusion for this topic.

L292-295: Invert the order of these sentences, it will facilitate reading avoiding misinterpretation.

L301: Please, add a conclusion for these paragraph and topic.

L303: Remove the commas.

L304: Add a dot after “energy”.

L310: Please, add a conclusion for this paragraph.

L318: Please, add a conclusion for these paragraph and topic.

L319: This topic needs to be presented in the Results section and then discussed here.

L324: “S3 Fig” should not be supplementary data and it must be presented in the body of the manuscript.

Conclusions

Good conclusion and in agreement with Results.

Figures

Fig 1 – OK

Fig 2 – merge Fig 2 (high nutrition PCA) with Fig S1 (low nutrition PCA), and present as A and B, respectively.

Fig 3 – merge Fig 3 (wet season PCA) with Fig S2 (dry season PCA), and present as A and B, respectively.

Fig 4 – merge Fig 4 with Fig 5, and present as A and B, respectively.

Fig 5 – use Fig S3, present the OJIP curves in the Results section.

Fig 6 – OK

Tables

Table 1 – use the S2 Table (firstly presented in the Methods section)

Table 2 – use the S1 Table

Table 3 – use the Table 1

Supplementary information

- Climatological data

- S3 Table (rename to S1 Table)

- S4 Table (rename to S2 Table)

- Raw data

Reviewer #2: Unfortunately, I must confess I do not have the expertise necessary to critically review the non-modulated fluorescence data presented in this manuscript. And, as the manuscript novelties and conclusions are heavily based on this type of analysis, I prefer to not evaluate the manuscript in order to not take any unfair decision. Anyway, a suggestion I gave to the authors is to improve the abstract to make it clear how based on non-modulated fluorescence this manuscript is. Had I known this beforehand, I would not have accepted to review the paper and saved the editor (and authors) some valuable time.

6. PLOS authors have the option to publish the peer review history of their article (what does this mean?). If published, this will include your full peer review and any attached files.

Reviewer #1: **Yes: **Vitor L. Nascimento

Reviewer #2: No

---

## [Author Response · Author response to Decision Letter 0]

16 Feb 2021

Changes were made throughout the manuscript and, in particular, in the discussion section to make the text and ideas more clear. We belief the manuscript has improved substantially. The hypotheses were reformulated to understand the effects of photochemical performance in different species. The figures were merged and presented as suggested.

More detail was given about the location and and climate of the region. Efforts were made to simplify the interpretation and presentation of data results. Supplementary figures were merged and included in the results and discussion sections. Supplementary Tables 1 and 2 were included in the main text. The discussion section was restructured to address: 1. Specific responses on chlorophyll fluorescence main variables under high-light conditions of degraded areas; 2. The effects of seasonality and fertilization treatments on photochemical performance; and 3. How fluorescence variables can be used as proxy for photosynthesis and growth determination. As wisely suggested concise conclusions and a brief summary were used in each topic of the discussion section with significant improve to the meaning of the text.

---

## [Decision Letter · Decision Letter 1]

12 Mar 2021

PONE-D-20-35817R1

Fertilization and seasonality influence on the photochemical performance of leguminous tree species during Amazonian reforestation

PLOS ONE

Dear Dr. Jaquetti,

Thank you for submitting your manuscript to PLOS ONE. After careful consideration, we feel that it has merit but does not fully meet PLOS ONE’s publication criteria as it currently stands. Therefore, we invite you to submit a revised version of the manuscript that addresses the points raised during the review process.

Your manuscript has been read and commented on by one independent reviewer who are experts in the field and have previously reviewed your mentioned. As the reviewer have previously mentioned, there is no doubt about the importance of the topic of your choice.

Reviewer has also suggested minor improvements to the main text and legends. I am therefore returning the manuscript to you for minor revisions and believe that pending these revisions your article will be acceptable for publication.

We look forward to receiving your revised manuscript.

All my best wishes,

Wagner

-- 

Prof. Wagner L. Araújo

Departamento de Biologia Vegetal 

Universidade Federal de Viçosa 

36570-900 Viçosa, MG, Brazil 

Tel: +55 (31) 3612.5358 

E-mail: wlaraujo@ufv.br

Academic Editor PLOS ONE

Journal Requirements:

Reviewers' comments:

Reviewer's Responses to Questions

**Comments to the Author**

1. If the authors have adequately addressed your comments raised in a previous round of review and you feel that this manuscript is now acceptable for publication, you may indicate that here to bypass the “Comments to the Author” section, enter your conflict of interest statement in the “Confidential to Editor” section, and submit your "Accept" recommendation.

Reviewer #1: (No Response)

2. Is the manuscript technically sound, and do the data support the conclusions?

Reviewer #1: Yes

3. Has the statistical analysis been performed appropriately and rigorously? 

Reviewer #1: Yes

4. Have the authors made all data underlying the findings in their manuscript fully available?

Reviewer #1: Yes

5. Is the manuscript presented in an intelligible fashion and written in standard English?

Reviewer #1: Yes

6. Review Comments to the Author

Reviewer #1: The authors improved the manuscript, and the message is much clearer now. However, the work still needs to be changed to be fully accepted. Below are my new comments.

Clitoria, Inga and Acacia are genus, not species. The same for Cenostigma, Senna and Dipteryx. Please correct this in all manuscript.

If authors cannot confirm the species used in the experiments, add "spp." after the genus name in all text, figures, and tables. In the previous version of the manuscript, the way that the species name was presented in the figure 1 captions was correct, now the authors only present the name of the genus.

Moreover, please standardize the use of "N" or "nitrogen" in the text. Remember that once the chemical element symbol is displayed, there is no need to write its name again.

7. PLOS authors have the option to publish the peer review history of their article (what does this mean?). If published, this will include your full peer review and any attached files.

Reviewer #1: **Yes: **Vitor L Nascimento

---

## [Author Response · Author response to Decision Letter 1]

10 Apr 2021

Responses to Reviewer #1:

The authors improved the manuscript, and the message is much clearer now. However, the work still needs to be changed to be fully accepted. Below are my new comments.

Authors' response: The authors are much appreciated with your revisions and comments. All suggestions were incorporated with much improve to the manuscript significantly. 

Clitoria, Inga and Acacia are genus, not species. The same for Cenostigma, Senna and Dipteryx. Please correct this in all manuscript. If authors cannot confirm the species used in the experiments, add "spp." after the genus name in all text, figures, and tables. In the previous version of the manuscript, the way that the species name was presented in the figure 1 captions was correct, now the authors only present the name of the genus.

Authors' response: The name of species were changed as previously presented. Acacia was not confirmed to species level, therefore spp. was used. 

Moreover, please standardize the use of "N" or "nitrogen" in the text. Remember that once the chemical element symbol is displayed, there is no need to write its name again.

Authors' response: As suggested the abbreviation of chemical element symbols were standardize and revised in the main text.

---

## [Editor Report · Decision Letter 2]

7 May 2021

Fertilization and seasonality influence on the photochemical performance of tree legumes in forest plantation for area recovery in the Amazon

PONE-D-20-35817R2

Dear Dr. Jaquetti,

We’re pleased to inform you that your manuscript has been judged scientifically suitable for publication and will be formally accepted for publication once it meets all outstanding technical requirements.

Kind regards,

Wagner L. Araujo, Ph.D

Academic Editor

PLOS ONE

Additional Editor Comments (optional):

Thank you very much for clearly answering all comments raised in previous round of review.

Reviewers' comments:

All comments have been addressed.

---

## [Editor Report · Acceptance letter]

14 May 2021

PONE-D-20-35817R2 

Fertilization and seasonality influence on the photochemical performance of tree legumes in forest plantation for area recovery in the Amazon 

Dear Dr. Jaquetti:

I'm pleased to inform you that your manuscript has been deemed suitable for publication in PLOS ONE. Congratulations! Your manuscript is now with our production department. 

Kind regards, 

on behalf of

Dr. Wagner L. Araujo 

Academic Editor

PLOS ONE